# FALCON: FAST AND LIGHTWEIGHT CONVOLUTION FOR COMPRESSING AND ACCELERATING CNN

## ABSTRACT

How can we efficiently compress Convolutional Neural Networks (CNN) while retaining their accuracy on classification tasks? A promising direction is based on depthwise separable convolution which replaces a standard convolution with a depthwise convolution and a pointwise convolution. However, previous works based on depthwise separable convolution are limited since 1) they are mostly heuristic approaches without a precise understanding of their relations to standard convolution, and 2) their accuracies do not match that of the standard convolution.

In this paper, we propose FALCON, an accurate and lightweight method for compressing CNN. FALCON is derived by interpreting existing convolution methods based on depthwise separable convolution using EHP, our proposed mathematical formulation to approximate the standard convolution kernel. Such interpretation leads to developing a generalized version rank-$k$ FALCON which further improves the accuracy while sacrificing a bit of compression and computation reduction rates. In addition, we propose FALCON-branch by fitting FALCON into the previous state-of-the-art convolution unit ShuffleUnitV2 which gives even better accuracy. Experiments show that FALCON and FALCON-branch outperform 1) existing methods based on depthwise separable convolution and 2) standard CNN models by up to $8\times$ compression and $8\times$ computation reduction while ensuring similar accuracy. We also demonstrate that rank-$k$ FALCON provides even better accuracy than standard convolution in many cases, while using a smaller number of parameters and floating-point operations.

## 1 INTRODUCTION

How can we efficiently reduce size and energy consumption of Convolutional Neural Networks (CNN) while maintaining their accuracy on classification tasks? Nowadays, CNN is widely used in various areas including computer vision (Krizhevsky et al. (2012); Simonyan & Zisserman (2014); Szegedy et al. (2017)), natural language processing (Yin et al. (2016)), recommendation system (Kim et al. (2016a)), etc. In addition, model compression has become an important technique due to an increase in the model capacity and the number of parameters in CNN. One recent and promising direction for compressing CNN is depthwise separable convolution (Sifre (2014)) which replaces standard convolution with depthwise and pointwise convolutions. The depthwise convolution applies a separate 2D convolution kernel for each input channel, and the pointwise convolution changes the channel size using $1\times1$ convolution (details in Section 2.1). Several recent methods (Howard et al. (2017); Sandler et al. (2018); Zhang et al. (2017)) based on depthwise separable convolution show reasonable performances in terms of compression and computation reduction.

However, existing approaches based on depthwise separable convolution have several crucial limitations. First, they are heuristic methods, and their relation to the standard convolution is not clearly identified. Second, due to their heuristic nature, generalizing the methods is difficult. Third, although they give reasonable compression and computation reduction, their accuracy is not sufficient compared to that of standard-convolution-based models.

In this paper, we propose FALCON, an accurate and lightweight method for compressing CNN. FALCON overcomes the limitations of the previous methods based on the depthwise separable convolution using the following two main ideas. First, we precisely define the relationship between the standard convolution and the depthwise separable convolution using EHP (Extended Hadamard

Product), which is our proposed mathematical formulation to correlate the standard convolution kernel with the depthwise convolution kernel and the pointwise convolution kernel. We then design FALCON by fine-tuning and reordering the results of EHP to improve the accuracy of convolution operations. Second, based on the precise definition, we generalize the FALCON to design rank-$k$ FALCON, which further improves accuracy while sacrificing a bit of compression and computation reduction rates. We also propose FALCON-branch by fitting FALCON into the state-of-the-art convolution unit ShuffleUnitV2 which gives even higher accuracy. As a result, FALCON and FALCON-branch provide a superior accuracy compared to other methods based on depthwise separable convolution, with similar compression and computation reduction rates, and rank-$k$ FALCON further improves accuracy, outperforming even the original convolution in many cases. Our contributions are summarized as follows:

- **Generalization.** We analyze and generalize depthwise separable convolution to our proposed EHP (Extended Hadamard Product) operation. This generalization enables a precise understanding of the relationship between depthwise separable convolution and standard convolution. Furthermore, with fine-tuning operations, it leads to our proposed method FALCON.
- **Algorithm.** We propose FALCON, a CNN compression method based on depthwise separable convolution. FALCON is carefully designed to compress CNN with little accuracy loss. We also propose rank-$k$ FALCON to further improve the accuracy with a little sacrifice in compression and computation reduction rates. FALCON can be easily integrated into other architectures, and we propose FALCON-branch which combines FALCON with a branch architecture for a better performance. We theoretically analyze the compression and computation reduction rates of FALCON and other competitors.
- **Experiments.** We perform extensive experiments and show that FALCON 1) outperforms other state-of-the-art methods based on depthwise separable convolution for compressing CNN, and 2) provides up to $8\times$ compression and computation reduction compared to the standard convolution while giving similar accuracies. Furthermore, we show that rank-$k$ FALCON provides even better accuracy than the standard convolution in many cases while using a smaller number of parameters and floating-point operations.

The rest of this paper is organized as follows. Section 2 explains preliminaries. Section 3 describes our proposed method FALCON. Section 4 presents experimental results. After discussing related works in Section 5, we conclude in Section 6.

## 2 PRELIMINARY

We describe preliminaries on depthwise separable convolution and methods based on depthwise separable convolution. Symbols used in this paper are described in Table 4 of Appendix.

### 2.1 DEPTHWISE SEPARABLE CONVOLUTION

Depthwise Separable Convolution (DSConv) consists of two sub-layers: depthwise convolution and pointwise convolution. The architecture of each convolution layer in DSConv is illustrated in Figure 5(a). Depthwise convolution (DWConv) kernel consists of several $D \times D$ 2-dimensional filters. The number of 2-dimension filters is the same as that of input feature maps. Each filter is applied on the corresponding input feature map, and produces an output feature map. Pointwise convolution (PWConv), known as $1 \times 1$ convolution, is a standard convolution with kernel size 1.

DSConv is defined as follows:

$$\mathcal{O}'_{h',w',m} = \sum_{i=1}^{D}\sum_{j=1}^{D} \mathcal{D}_{i,j,m} \cdot \mathcal{I}_{h_i,w_j,m} \tag{1}$$

$$\mathcal{O}_{h',w',n} = \sum_{m=1}^{M} \mathbf{P}_{m,n} \cdot \mathcal{O}'_{h',w',m} \tag{2}$$

where $\mathcal{D}_{i,j,m}$ and $\mathbf{P}_{m,n}$ are depthwise convolution kernel and pointwise convolution kernel, respectively. $\mathcal{O}'_{h',w',m} \in \mathbb{R}^{H' \times W' \times M}$ denotes intermediate feature maps. DSConv performs DWConv on

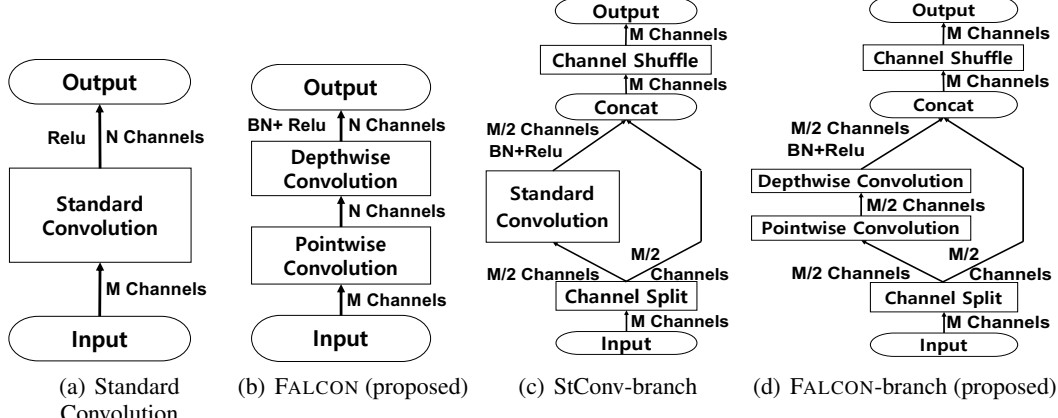

Figure 1: Comparison of architectures. BN denotes batch-normalization. Relu and Relu6 are activation functions. (a) Standard convolution. (b) Our proposed FALCON. (c) Standard convolution with branch (StConv-branch). (d) FALCON-branch which combines FALCON with StConv-branch.

input feature maps $\mathfrak{I}_{h_i, w_j, m}$ using equation 1, and generates intermediate feature maps $\mathcal{O}'_{h', w', m}$. Then, DSConv performs PWConv on $\mathcal{O}'_{h', w', m}$ using equation 2, and generates output feature maps $\mathcal{O}_{h', w', n}$.

## 2.2 METHODS BASED ON DEPTHWISE SEPARABLE CONVOLUTION

Several CNN methods have been proposed based on Depthwise Separable Convolution (DSConv) recently. DSConv was first introduced by Sifre (2014). Chollet & Franois (2016) built Xception module using DSConv in a few layers. Howard et al. (2017) built Mobilenet with all convolution layers replaced by DSConv. Sandler et al. (2018) built MobileNetV2 with inverted bottleneck block, denoted as MobileConvV2 in this paper. Zhang et al. (2017) built a CNN model with Shufflenet Unit, denoted as ShuffleUnit in this paper. Ma et al. (2018) improved Shufflenet by designing ShuflenetV2 Unit, denoted as ShuffleUnitV2 in this paper. The architecture and detailed descriptions are in Figure 5 and Appendix D.

## 3 PROPOSED METHOD

We describe FALCON, our proposed method for compressing CNN. We first define Extended Hadamard Product (EHP), a key mathematical formulation to generalize depthwise separable convolution, in Section 3.1. We interpret depthwise separable convolution used in Mobilenet using EHP in Section 3.2. We propose FALCON in Section 3.3 and explain why FALCON can replace standard convolution. Then, we propose rank-k FALCON, which extends the basic FALCON, in Section 3.4. We show that FALCON can be easily integrated into a branch architecture to compress it with little sacrifice of accuracy in Section 3.5. Finally, we theoretically analyze the performance of FALCON in Appendix C.

### 3.1 EXTENDED HADAMARD PRODUCT (EHP)

We define Extended Hadamard Product (EHP), a generalized elementwise product for two operands of different shapes, to generalize the formulation of the relation between standard convolution and depthwise separable convolution. Before generalizing the formulation, we give an example of formulating the relation between standard convolution and depthwise separable convolution. Suppose we have a 4-order standard convolution kernel $\mathcal{K} \in \mathbb{R}^{I \times J \times M \times N}$, a 3-order depthwise convolution kernel $\mathcal{D} \in \mathbb{R}^{I \times J \times M}$, and a pointwise convolution kernel $\mathbf{P} \in \mathbb{R}^{M \times N}$. Let $\mathcal{K}_{i,j,m,n}$ be $(i, j, m, n)$-th element of $\mathcal{K}$, $\mathcal{D}_{i,j,m}$ be $(i, j, m)$-th element of $\mathcal{D}$, and $\mathbf{P}_{m,n}$ be $(m, n)$-th element of $\mathbf{P}$. Then, it can be shown that applying depthwise convolution with $\mathcal{D}$ and pointwise convolution with $\mathbf{P}$ is equivalent to applying standard convolution kernel $\mathcal{K}$ where $\mathcal{K}_{i,j,m,n} = \mathcal{D}_{i,j,m} \cdot \mathbf{P}_{m,n}$ (see Section 3.2 for detailed proof).

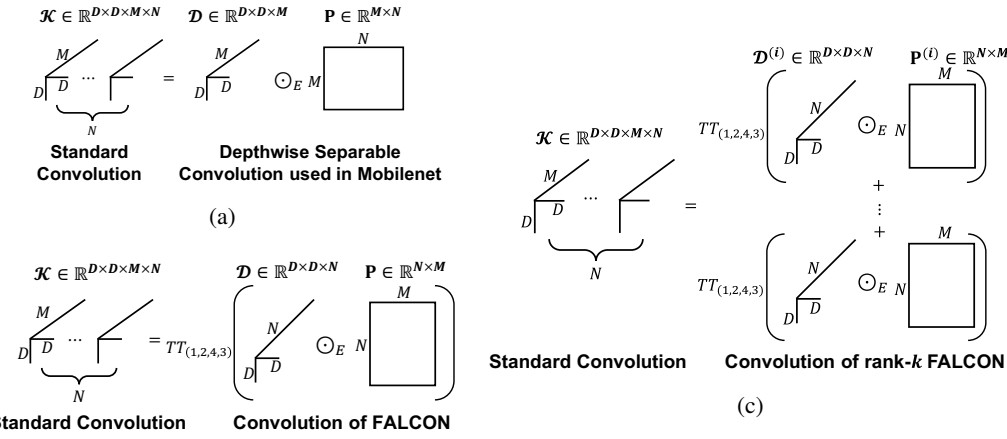

Figure 2: (a) Relation between standard convolution and depthwise separable convolution expressed with EHP. The common axes correspond to the input channel-axis of standard convolution. (b) Relation between standard convolution and FALCON expressed with EHP. The common axes correspond to the output channel-axis of standard convolution. $TT_{(1,2,4,3)}$ indicates tensor transpose operation to permute the third and the fourth dimensions of a tensor. (c) Relation between the standard convolution and rank-$k$ FALCON expressed with EHP.

To formally express this relation, we define Extended Hadamard Product (EHP) as follows.

**Definition 1** (Extended Hadamard Product). Given $p$-order tensor $\mathcal{D} \in \mathbb{R}^{I_1 \times \cdots \times I_{p-1} \times M}$ and $q$-order tensor $\mathcal{P} \in \mathbb{R}^{M \times J_1 \times \cdots \times J_{q-1}}$, the Extended Hadamard Product $\mathcal{D} \odot_E \mathcal{P}$ of $\mathcal{D}$ and $\mathcal{P}$ is defined to be the tensor $\mathcal{K} \in \mathbb{R}^{I_1 \times \cdots \times I_{p-1} \times M \times J_1 \times \cdots \times J_{q-1}}$ where the last axis of $\mathcal{D}$ and the first axis of $\mathcal{P}$ are the common axes such that

$$\mathcal{K}_{i_1,\ldots,i_{p-1},m,j_1,\ldots,j_{q-1}} = \mathcal{D}_{i_1,\ldots,i_{p-1},m} \cdot \mathcal{P}_{m,j_1,\ldots,j_{q-1}}$$

for all elements of $\mathcal{K}$. □

Contrary to Hadamard Product which is defined only if the shapes of the two operands are the same, Extended Hadamard Product (EHP) deals with tensors of different shapes. Now, we define a special case of Extended Hadamard Product (EHP) for a third-order tensor and a matrix.

**Definition 2** (Extended Hadamard Product for a third order tensor and a matrix). Given a third-order tensor $\mathcal{D} \in \mathbb{R}^{I \times J \times M}$ and a matrix $\mathbf{P} \in \mathbb{R}^{M \times N}$, the Extended Hadamard Product $\mathcal{D} \odot_E \mathbf{P}$ is defined to be the tensor $\mathcal{K} \in \mathbb{R}^{I \times J \times M \times N}$ where the third axis of the tensor $\mathcal{D}$ and the first axis of the matrix $\mathbf{P}$ are the common axes such that

$$\mathcal{K}_{i,j,m,n} = \mathcal{D}_{i,j,m} \cdot \mathbf{P}_{m,n}.$$

for all elements of $\mathcal{K}$. □

We will see that the depthwise separable convolution in Mobilenet can be easily expressed with EHP in Section 3.2; we also propose a new architecture FALCON based on EHP in Section 3.3. EHP is also a core operation that helps us understand other convolution architectures including MobilenetV2 and Shufflenet (see Appendix E).

## 3.2 DEPTHWISE SEPARABLE CONVOLUTION AND EHP

In this section, we discuss how to represent the convolution layer of Mobilenet as Extended Hadamard Product (EHP) described in Section 3.1. We interpret the depthwise separable convolution, which is the convolution of Mobilenet, as an application of EHP. This interpretation leads to designing a better convolution architecture FALCON in Section 3.3.

We represent the relationship between standard convolution kernel $\mathcal{K} \in \mathbb{R}^{D \times D \times M \times N}$ and depthwise separable convolution consisting of depthwise convolution kernel $\mathcal{D} \in \mathbb{R}^{D \times D \times M}$ and pointwise convolution kernel $\mathbf{P} \in \mathbb{R}^{M \times N}$ using one EHP operation. Figure 2(a) illustrates the relationship between standard convolution and depthwise separable convolution used in Mobilenet. We show that applying depthwise separable convolution with $\mathcal{D}$ and $\mathbf{P}$ is equivalent to applying standard convolution with a kernel $\mathcal{K}$ which is constructed from $\mathcal{D}$ and $\mathbf{P}$.

**Theorem 1.** *Applying depthwise separable convolution with depthwise convolution kernel* $\mathcal{D} \in \mathbb{R}^{D \times D \times M}$ *and pointwise convolution kernel* $\mathbf{P} \in \mathbb{R}^{M \times N}$ *is equivalent to applying standard convolution with kernel* $\mathcal{K} = \mathcal{D} \odot_E \mathbf{P}$.

*Proof.* See Appendix B.1. □

### 3.3 FAST AND LIGHTWEIGHT CONVOLUTION (FALCON)

We propose FALCON (FAst and Lightweight CONvolution), a novel lightweight convolution that replaces standard convolution. FALCON is an efficient method with fewer parameters and computations than those that the standard convolution requires. In addition, FALCON has better accuracy than competitors while having similar compression and computation reduction rates. The main idea of FALCON is 1) to carefully align depthwise and pointwise convolutions, and 2) initialize kernels using the convolution kernels of the trained standard model. We observe that a typical convolution has more output channels than input channels. In such a setting, performing depthwise convolution after pointwise convolution would allow the depthwise convolution to extract more features from richer feature space; on the other hand, performing pointwise convolution after depthwise convolution as in Mobilenet only combines features extracted from a limited feature space. Based on the observation, FALCON first applies pointwise convolution to generate an intermediate tensor $\mathcal{O}' \in \mathbb{R}^{H \times W \times N}$ and then applies depthwise convolution.

We represent the relationship between standard convolution kernel $\mathcal{K} \in \mathbb{R}^{D \times D \times M \times N}$ and FALCON by applying an EHP operation on pointwise convolution kernel $\mathbf{P} \in \mathbb{R}^{N \times M}$ and depthwise convolution kernel $\mathcal{D} \in \mathbb{R}^{D \times D \times N}$ in Figure 2(b). In FALCON, the kernel $\mathcal{K}$ is represented by EHP of $\mathcal{D}$ and $\mathbf{P}$ as follows:

$$\mathcal{K} = TT_{(1,2,4,3)}(\mathcal{D} \odot_E \mathbf{P}) \quad \text{s.t.} \quad \mathcal{K}_{i,j,m,n} = \mathbf{P}_{n,m} \cdot \mathcal{D}_{i,j,n}.$$

where $TT_{(1,2,4,3)}$ indicates tensor transpose operation to permute the third and the fourth dimensions of a tensor. Note that the common axis is the output channel axis of the standard convolution, unlike EHP for depthwise separable convolution where the common axis is the input channel axis of the standard convolution.

As in Section 3.2, we show that applying FALCON is equivalent to applying standard convolution with a specially constructed kernel.

**Theorem 2.** FALCON *which applies pointwise convolution with kernel* $\mathbf{P} \in \mathbb{R}^{N \times M}$ *and then depthwise convolution with kernel* $\mathcal{D} \in \mathbb{R}^{D \times D \times N}$ *is equivalent to applying standard convolution with kernel* $\mathcal{K} = TT_{(1,2,4,3)}(\mathcal{D} \odot_E \mathbf{P})$.

*Proof.* See Appendix B.2. □

Based on the equivalence, we initialize pointwise convolution and depthwise convolution kernels $\mathcal{D}$ and $\mathbf{P}$ of FALCON by fitting them to the convolution kernels of the trained standard model; i.e., $\mathcal{D}, \mathbf{P} = \arg\min_{\mathcal{D}',\mathbf{P}'} ||\mathcal{K} - TT_{(1,2,4,3)}(\mathcal{D}' \odot_E \mathbf{P}')||_F$. After pointwise convolution and depthwise convolution, we add batch-normalization and ReLU activation function as shown in Figure 1(b). We note that FALCON significantly reduces the numbers of parameters and FLOPs compared to standard convolution, which we discuss at Appendix C.

### 3.4 RANK-$k$ FALCON

We propose rank-$k$ FALCON, an extended version of FALCON that improves accuracy while sacrificing a bit of compression and computation reduction rates. The main idea is to perform $k$ independent FALCON operations and sum up the result. Then, we apply batch-normalization (BN) and ReLU activation function to the summed result. Since each FALCON operation requires independent parameters for pointwise convolution and depthwise convolution, the number of parameters increases and thus the compression and the computation reduction rates decrease; however, it improves accuracy by enlarging the model capacity. We formally define the rank-$k$ FALCON with EHP as follows.

**Definition 3** (Rank-$k$ FALCON with Extended Hadamard Product). *Rank-$k$ FALCON expresses standard convolution kernel* $\mathcal{K} \in \mathbb{R}^{D \times D \times M \times N}$ *as EHP of depthwise convolution kernel* $\mathcal{D}^{(i)} \in$

$\mathbb{R}^{D \times D \times N}$ *and pointwise convolution kernel* $\mathbf{P}^{(i)} \in \mathbb{R}^{N \times M}$ *for* $i = 1, 2, ..., k$ *such that*

$$\boldsymbol{\mathcal{K}} = \sum_{i=1}^{k} TT_{(1,2,4,3)}(\boldsymbol{\mathfrak{D}}^{(i)} \odot_E \mathbf{P}^{(i)}) \qquad s.t. \qquad \boldsymbol{\mathcal{K}}_{i,j,m,n} = \sum_{i=1}^{k} \mathbf{P}_{n,m}^{(i)} \cdot \boldsymbol{\mathfrak{D}}_{i,j,n}^{(i)}$$

□

Figure 2(c) illustrates the relation between standard convolution and rank-$k$ FALCON. For each $i = 1, 2, ..., k$, we construct the tensor $\boldsymbol{\mathcal{K}}^{(i)}$ using EHP of the depthwise convolution kernel $\boldsymbol{\mathfrak{D}}^{(i)}$ and the pointwise convolution kernel $\mathbf{P}^{(i)}$. Then, we construct the standard kernel $\boldsymbol{\mathcal{K}}$ by the element-wise sum of the tensors $\boldsymbol{\mathcal{K}}^{(i)}$ for all $i$.

## 3.5 FALCON-BRANCH

FALCON can be easily integrated into a CNN architecture called standard convolution operation with a branch (StConv-branch), which consists of two branches: standard convolution on the left branch and a residual connection on the right branch (see Figure 1(c)). Ma et al. (2018) improved the performance of CNN by applying depthwise and pointwise convolutions on the left branch of StConv-branch. Since FALCON replaces standard convolution, we observe that StConv-branch can be easily compressed by applying FALCON on the left branch.

StConv-branch first splits an input in half along the depth dimension. A standard convolution operation is applied to one half, and no operation to the other half. The two are concatenated along the depth dimension, and an output is produced by shuffling the channels of the concatenated tensor. FALCON-branch (see Figure 1(d)) is constructed by replacing the standard convolution branch (left branch) of StConv-branch with FALCON. Advantages of FALCON-branch are that 1) the branch architecture improves the efficiency since convolutions are applied to only half of input feature maps and that 2) FALCON further compresses the left branch effectively. FALCON-branch is initialized by fitting FALCON to the standard convolution kernel of the left branch of StConv-branch.

## 4 EXPERIMENTS

We validate the performance of FALCON through extensive experiments. We aim to answer the following questions:

- **Q1. Accuracy vs. Compression (Section 4.3).** What are the accuracy and the compression tradeoffs of FALCON, FALCON-branch, and competitors? Which method gives the best accuracy for a given compression rate?
- **Q2. Accuracy vs. Computation (Section 4.4).** What are the accuracy and the computation tradeoffs of FALCON, FALCON-branch, and competitors? Which method gives the best accuracy for a given amount of computation?
- **Q3. Rank-$k$ FALCON (Section 4.5).** How do the accuracy, the number of parameters, and the number of FLOPs change as the rank $k$ increases in FALCON?

## 4.1 EXPERIMENTAL SETUP

**Datasets.** We perform image classification task on four famous datasets - CIFAR10, CIFAR100, SVHN, and ImageNet. Detailed information of these datasets is described in Table 1.

Table 1: Datasets.

| dataset | # of classes | input size | # of train | # of test |
|---------|--------------|------------|------------|-----------|
| CIFAR-10[1] | 10 | $32 \times 32 \times 3$ | $10 \times 6000$ | 10000 |
| CIFAR-100[2] | 100 | $32 \times 32 \times 3$ | $100 \times 600$ | 10000 |
| SVHN[3] | 10 | $32 \times 32$ | 73257 | 26032 |
| ImageNet[4] | 1000 | $224 \times 224 \times 3$ | $1.2 \times 10^6$ | 150000 |

[1]`https://www.cs.toronto.edu/~kriz/cifar.html`
[2]`https://www.cs.toronto.edu/~kriz/cifar.html`
[3]`http://ufldl.stanford.edu/housenumbers/`
[4]`http://www.image-net.org`

**Models.** For CIFAR10, CIFAR100, and SVHN datasets, we choose VGG19 and ResNet34 to evaluate the performance. We shrink the sizes of both models since the sizes of these three datasets are smaller than that of Imagenet. In VGG19, we reduce the number of fully connected layers and the number of features in fully connected layers: three large fully connected layers (4096-4096-1000) in VGG19 are replaced with two small fully connected layers (512-10 or 512-100). In ResNet34, we remove the first $7 \times 7$ convolution layer and max-pooling layer since the input size ($32 \times 32$) of these datasets is smaller than the input size ($224 \times 224$) of ImageNet. On both models, we replace all standard convolution layers (except for the first convolution layer) with those of FALCON or other competitors in order to compress and accelerate the model. For ImageNet, we choose VGG16_BN (VGG16 with batch normalization after every convolution layer) and ResNet18. We use the pretrained model from Pytorch model zoo as the baseline model with standard convolution, and replace the standard convolution with other types of convolutions.

**Competitors.** We compare FALCON and FALCON-branch with four convolution units consisting of depthwise convolution and pointwise convolution: DSConv, MobileConvV2, ShuffleUnit, and ShuffleUnitV2 (see Figure 5, Section 2.1, and Appendix D for more details). To evaluate the effectiveness of fitting depthwise and pointwise convolution kernels to standard convolution kernel, we build EHP-in which is DSConv where kernels $\mathcal{D}$ and $\mathbf{P}$ are fitted from the pretrained standard convolution kernel $\mathcal{K}$; i.e., $\mathcal{D}, \mathbf{P} = \arg\min_{\mathcal{D}', \mathbf{P}'} ||\mathcal{K} - \mathcal{D}' \odot_E \mathbf{P}'||_F$.

**Implementation.** We construct all models using Pytorch framework. All the models are trained and tested on GeForce GTX 1080 Ti GPU.

## 4.2 FITTING CONVOLUTION UNIT INTO MODEL

We evaluate the performance of FALCON against DSConv, MobileConvV2, ShuffleUnit, and ShuffleUnitV2. We take each standard convolution layer (StConv) as a unit, and replace StConv with those from FALCON or other competitors. We evaluate the classification accuracy, the number of parameters in the model, and the number of FLOPs needed for forwarding one image. We only explain how to apply FALCON in this section. The details of how to fit other convolution units into the models are described in Appendix F.

**FALCON.** When replacing StConv with FALCON, we use the same setting as that of StConv. I.e., if there are BN and ReLU after StConv, we add BN and ReLU at the end of FALCON; if there is only ReLU after StConv, we add only ReLU at the end of FALCON. This is because FALCON is initialized by approximating the StConv kernel using EHP. Using the same setting for BN and ReLU as StConv is more efficient for FALCON to approximate the StConv. We initialize the pointwise convolution kernel and the depthwise convolution kernel of FALCON by approximating the pretrained standard convolution kernel using EHP. The approximation process is as follows: 1) we first initialize the pointwise convolution kernel and the depthwise convolution kernel randomly, and 2) the pointwise convolution kernel and the depthwise convolution kernel are updated using gradient descent such that the mean squared error of their EHP product and the standard convolution kernel is minimized. Rank-$k$ FALCON uses the same initialization method.

## 4.3 ACCURACY VS. COMPRESSION

We evaluate the accuracy and the compression rate of FALCON and competitors. Table 2 shows the results on four image datasets. Note that FALCON or FALCON-branch provides the highest accuracy in 7 out of 8 cases while using similar or smaller number of parameters than competitors. Specifically, FALCON and FALCON-branch achieve up to $8\times$ compression rates with less than $1\%$ accuracy drop compared to that of the standard convolution (StConv). Figure 3 shows the tradeoff between accuracy and the number of parameters. Note that FALCON and FALCON-branch show the best tradeoff (closest to the "best" point) between accuracy and compression rate, giving the highest accuracy with similar compression rates.

**Ablation study.** We perform an ablation study on two components: 1) the order of depthwise and pointwise convolutions, and 2) initialization. We observe that with a similar number of parameters, 1) FALCON and FALCON without initialization always result in better accuracy than EHP-in and

Table 2: FALCON and FALCON-branch gives the best accuracy for similar number of parameters and FLOPs. Bold font indicates the best accuracy among competing compression methods.

(a) VGG19-CIFAR10

| ConvType | Accuracy | # of param | # of FLOPs |
|---|---|---|---|
| StConv | 93.56% | 20.30M | 398.70M |
| FALCON | **93.40%** | 2.56M | 47.23M |
| FALCON without initialization | 92.88% | 2.56M | 47.23M |
| FALCON-branch 1.75× | 93.14% | 2.64M | 54.17M |
| EHP-in | 92.06% | 2.56M | 46.41M |
| DSC | 91.54% | 2.56M | 48.02M |
| MobileConvV2-0.5 | 92.65% | 2.67M | 51.80M |
| ShuffleUnit 2×(g=2) | 92.75% | 2.74M | 46.66M |
| ShuffleUnitV2 1.375× | 92.78% | 2.86M | 58.24M |

(b) ResNet34-CIFAR10

| ConvType | Accuracy | # of param | # of FLOPs |
|---|---|---|---|
| StConv | 94.01% | 21.29M | 292.52M |
| FALCON | **92.78%** | 2.63M | 46.33M |
| FALCON without initialization | 92.60% | 2.63M | 46.33M |
| FALCON-branch 1.625× | 92.64% | 2.47M | 58.86M |
| EHP-in | 91.73% | 2.62M | 38.41M |
| DSC | 91.54% | 2.62M | 38.41M |
| MobileConvV2-0.5 | 91.34% | 2.55M | 39.78M |
| ShuffleUnit 2×(g=2) | 91.74% | 3.08M | 49.78M |
| ShuffleUnit V2 1.375× | 92.16% | 2.98M | 51.30M |

(c) VGG19-CIFAR100

| ConvType | Accuracy | # of param | # of FLOPs |
|---|---|---|---|
| StConv | 72.10% | 20.35M | 398.75M |
| FALCON | 71.63% | 2.61M | 47.28M |
| FALCON without initialization | 71.80% | 2.61M | 47.28M |
| FALCON-branch 1.75× | **73.05%** | 2.68M | 54.21M |
| EHP-in | 68.29% | 2.61M | 46.46M |
| DSC | 68.18% | 2.61M | 48.07M |
| MobileConvV2-0.5 | 72.50% | 2.71M | 51.85M |
| ShuffleUnit 2×(g=2) | 72.73% | 2.79M | 46.71M |
| ShuffleUnit V2 1.375× | 72.32% | 2.91M | 58.29M |

(d) ResNet34-CIFAR100

| ConvType | Accuracy | # of param | # of FLOPs |
|---|---|---|---|
| StConv | 73.94% | 21.34M | 292.57M |
| FALCON | **71.83%** | 2.67M | 46.38M |
| FALCON without initialization | 71.80% | 2.67M | 46.38M |
| FALCON-branch 1.625× | 70.26% | 2.54M | 58.93M |
| EHP-in | 66.88% | 2.67M | 38.45M |
| DSC | 66.30% | 2.67M | 38.45M |
| MobileConvV2-0.5 | 65.00% | 2.59M | 39.83M |
| ShuffleUnit 2×(g=2) | 68.97% | 3.17M | 49.88M |
| ShuffleUnit V2 1.375× | 67.38% | 3.04M | 51.36M |

(e) VGG19-SVHN

| ConvType | Accuracy | # of param | # of FLOPs |
|---|---|---|---|
| StConv | 95.28% | 20.30M | 398.70M |
| FALCON | **95.45%** | 2.56M | 47.23M |
| FALCON without initialization | 94.51% | 2.56M | 47.23M |
| FALCON-branch 1.75× | 94.56% | 2.64M | 54.17M |
| EHP-in | 94.99% | 2.56M | 46.41M |
| DSC | 94.37% | 2.56M | 48.02M |
| MobileConvV2-0.5 | 93.28% | 2.67M | 51.80M |
| ShuffleUnit 2×(g=2) | 93.15% | 2.74M | 46.66M |
| ShuffleUnit V2 1.375× | 94.36% | 2.86M | 58.24M |

(f) ResNet34-SVHN

| ConvType | Accuracy | # of param | # of FLOPs |
|---|---|---|---|
| StConv | 94.83% | 21.29M | 292.52M |
| FALCON | 94.83% | 2.63M | 46.33M |
| FALCON without initialization | 94.75% | 2.63M | 46.33M |
| FALCON-branch 1.625× | **94.98%** | 2.47M | 58.86M |
| EHP-in | 94.06% | 2.62M | 38.41M |
| DSC | 94.03% | 2.62M | 38.41M |
| MobileConvV2-0.5 | 93.16% | 2.55M | 39.78M |
| ShuffleUnit 2×(g=2) | 93.68% | 3.08M | 49.78M |
| ShuffleUnit V2 1.375× | 94.35% | 2.98M | 51.30M |

(g) VGG16_BN-ImageNet

| ConvType | Top-1 Accuracy | Top-5 Accuracy | # of param | # of FLOPs |
|---|---|---|---|---|
| StConv | 73.37% | 91.50% | 138.37M | 15484.82M |
| FALCON | 71.63% | **90.57%** | 125.33M | 1950.75M |
| FALCON without initialization | **71.65%** | 90.47% | 125.33M | 1950.75M |
| FALCON-branch 1.5× | 68.24% | 88.51% | 125.30M | 1898.39M |
| EHP-in | 70.98% | 90.19% | 125.33M | 1910.56M |
| DSC | 70.34% | 89.71% | 125.33M | 1989.49M |
| MobileConvV2-0.5 | 67.80% | 87.90% | 125.44M | 2180.49M |
| ShuffleUnit 2×(g=2) | 70.40% | 89.84% | 125.77M | 2014.73M |
| ShuffleUnitV2 1.25× | 71.34% | 90.34% | 125.57M | 2180.65M |

(h) ResNet18-ImageNet

| ConvType | Top-1 Accuracy | Top-5 Accuracy | # of param | # of FLOPs |
|---|---|---|---|---|
| StConv | 69.76% | 89.08% | 11.69M | 1814.07M |
| FALCON | **66.64%** | 87.09% | 1.97M | 395.40M |
| FALCON without initialization | 66.19% | 86.86% | 1.97M | 395.40M |
| FALCON-branch 1.375× | 64.01% | 85.16% | 1.91M | 434.44M |
| EHP-in | 66.21% | 86.93% | 1.96M | 336.81M |
| DSC | 65.30% | 86.30% | 1.96M | 336.81M |
| MobileConvV2-0.5 | 58.99% | 81.55% | 1.90M | 340.06M |
| ShuffleUnit 2×(g=2) | 65.73% | 86.75% | 2.22M | 438.89M |
| ShuffleUnitV2 1.1875× | 66.29% | **87.32%** | 2.01M | 376.15M |

DSC, respectively, and 2) EHP-in always results in better accuracy than DSC. Furthermore, FALCON results in better accuracy than FALCON without initialization in 6 out of 8 cases. These observations prove our claims in Section 3.3 that 1) EHP-out (FALCON) is more efficient than EHP-in, and 2) the fitting and the initialization of kernels using EHP improves accuracy. Additionally, we observe that overall performance is more sensitive towards ordering compared to initialization.

## 4.4 ACCURACY VS. COMPUTATION

We evaluate the accuracy and the amount of computation of FALCON and competitors. We use the number of multiply-adds floating point operations (FLOPs) needed for forwarding one image to a model as the metric of computation. Table 2 also shows the accuracies and the number of FLOPs of methods on four image datasets. Note that FALCON or FALCON-branch provide the highest accuracy in 7 out of 8 cases while using similar FLOPs as competitors do. Compared to StConv, FALCON and FALCON-branch achieve up to 8× FLOPs reduction across different models on different datasets. Figure 4 shows the tradeoff between accuracy and the number of FLOPs. Note that FALCON and FALCON-branch show the best tradeoff (closest to the "best" point) between accuracy and computation, giving the highest accuracy with a similar number of FLOPs.

## 4.5 RANK-K FALCON

We evaluate the performance of rank-$k$ FALCON by increasing the rank $k$ and monitoring the changes in the numbers of parameters and FLOPs. In Table 3, we observe three trends as the rank $k$ increases: 1) the accuracy becomes higher than that of rank-1 FALCON, 2) the number of parameters increases,

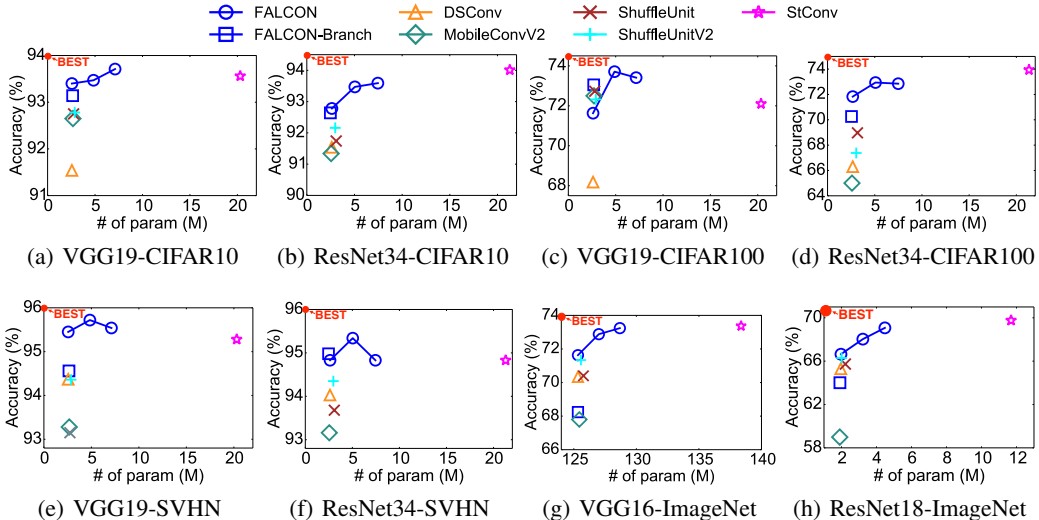

Figure 3: Accuracy w.r.t. number of parameters on different models and datasets. The three blue circles correspond to rank-1, 2, 3 FALCON (from left to right order), respectively. FALCON provides the best accuracy for a given number of parameters.

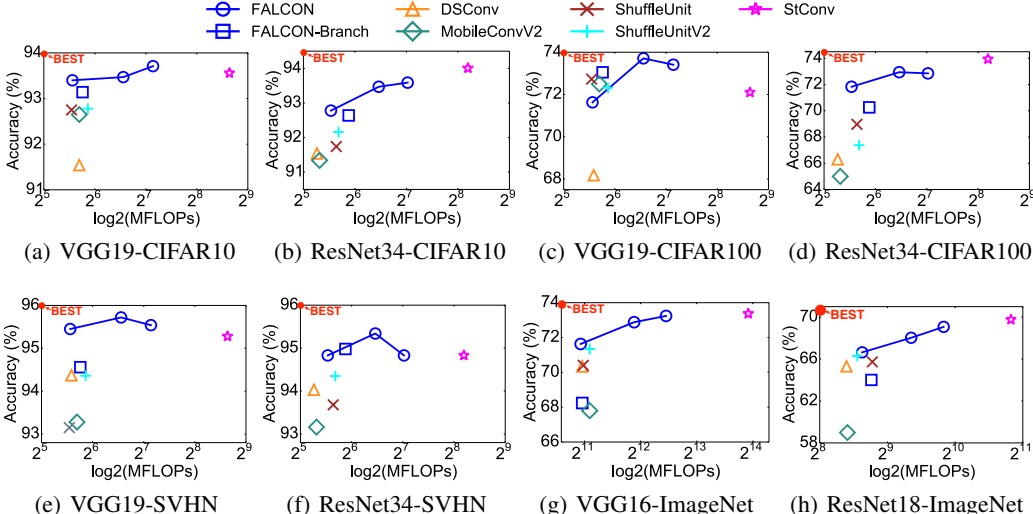

Figure 4: Accuracy w.r.t. FLOPs on different models and datasets. The three blue circles correspond to rank-1, 2, 3 FALCON (from left to right order), respectively. FALCON provides the best accuracy for a given number of FLOPs.

and 3) the number of floating point operations (FLOPs) increases. Although the rank $k$ that gives the best tradeoff of rank and compression/computation reduction varies, rank-$k$ FALCON improves the accuracy of FALCON in all cases. Especially, we note that rank FALCON often gives even higher accuracy than the standard convolution, while using smaller number of parameters and FLOPs. For example, rank-3 FALCON applied to VGG19 on CIFAR100 dataset shows 1.31 percentage point higher accuracy compared to the standard convolution, with $2.8\times$ smaller number of parameters and $2.8\times$ smaller number of FLOPs. Thus, rank-$k$ FALCON is a versatile method to further improve the accuracy of FALCON while sacrificing a bit of compression and computation.

## 5 RELATED WORK

Over the past several years, a lot of studies focused on compressing and accelerating DNN to reduce model size, running time, and energy consumption.

It is believed that DNNs are over-parameterized. Weight-sharing (Han et al. (2016); Ullrich et al. (2017); Chen et al. (2015); Choi et al. (2017); Agustsson et al. (2017)) is a common compression

Table 3: Rank-$k$ FALCON further improves accuracy while sacrificing a bit of compression and computation.

(a) VGG19-CIFAR10

| ConvType | Accuracy | # of param | | # of FLOPs | |
| --- | --- | --- | --- | --- | --- |
| StConv | 93.56% | 20.30M | | 398.70M | |
| FALCON-k1 | 93.40% | 2.56M | (7.93×) | 47.23M | (8.44×) |
| FALCON-k2 | 93.47% | 4.84M | (4.19×) | 94.20M | (4.23×) |
| FALCON-k3 | 93.71% | 7.11M | (2.86×) | 141.16M | (2.82×) |

(b) ResNet34-CIFAR10

| ConvType | Accuracy | # of param | | # of FLOPs | |
| --- | --- | --- | --- | --- | --- |
| StConv | 94.01% | 21.29M | | 292.52M | |
| FALCON-k1 | 92.78% | 2.63M | (8.10×) | 46.33M | (6.31×) |
| FALCON-k2 | 93.47% | 5.04M | (4.22×) | 88.21M | (3.32×) |
| FALCON-k3 | 93.59% | 7.45M | (2.86×) | 130.08M | (2.25×) |

(c) VGG19-CIFAR100

| ConvType | Accuracy | # of param | | # of FLOPs | |
| --- | --- | --- | --- | --- | --- |
| StConv | 72.10% | 20.35M | | 398.75M | |
| FALCON-k1 | 71.63% | 2.61M | (7.80×) | 47.28M | (8.43×) |
| FALCON-k2 | 73.71% | 4.88M | (4.17×) | 94.24M | (4.23×) |
| FALCON-k3 | 73.41% | 7.16M | (2.84×) | 141.21M | (2.82×) |

(d) ResNet34-CIFAR100

| ConvType | Accuracy | # of param | | # of FLOPs | |
| --- | --- | --- | --- | --- | --- |
| StConv | 73.94% | 21.34M | | 292.57M | |
| FALCON-k1 | 71.83% | 2.67M | (7.99×) | 46.38M | (6.31×) |
| FALCON-k2 | 72.94% | 5.08M | (4.20×) | 88.25M | (3.32×) |
| FALCON-k3 | 72.85% | 7.49M | (2.85×) | 130.13M | (2.25×) |

(e) VGG19-SVHN

| ConvType | Accuracy | # of param | | # of FLOPs | |
| --- | --- | --- | --- | --- | --- |
| StConv | 95.28% | 20.30M | | 398.70M | |
| FALCON-k1 | 95.45% | 2.56M | (7.93×) | 47.23M | (8.44×) |
| FALCON-k2 | 95.72% | 4.84M | (4.19×) | 94.20M | (4.23×) |
| FALCON-k3 | 95.54% | 7.11M | (2.86×) | 141.16M | (2.82×) |

(f) ResNet34-SVHN

| ConvType | Accuracy | # of param | | # of FLOPs | |
| --- | --- | --- | --- | --- | --- |
| StConv | 94.83% | 21.29M | | 292.52M | |
| FALCON-k1 | 94.83% | 2.63M | (8.10×) | 46.33M | (6.31×) |
| FALCON-k2 | 95.34% | 5.04M | (4.22×) | 88.21M | (3.32×) |
| FALCON-k3 | 94.83% | 7.45M | (2.86×) | 130.08M | (2.25×) |

(g) VGG16_BN-ImageNet

| ConvType | Top-1 Accuracy | Top-5 Accuracy | # of param | | # of FLOPs | |
| --- | --- | --- | --- | --- | --- | --- |
| StConv | 73.37% | 91.50% | 138.37M | | 15484.82M | |
| FALCON-k1 | 71.63% | 90.57% | 125.33M | (1.10×) | 1950.75M | (7.94×) |
| FALCON-k2 | 72.88% | 91.19% | 127.00M | (1.09×) | 3777.86M | (4.10×) |
| FALCON-k3 | 73.24% | 91.54% | 128.68M | (1.08×) | 5604.97M | (2.76×) |

(h) ResNet18-ImageNet

| ConvType | Top-1 Accuracy | Top-5 Accuracy | # of param | | # of FLOPs | |
| --- | --- | --- | --- | --- | --- | --- |
| StConv | 69.76% | 89.08% | 11.69M | | 1814.07M | |
| FALCON-k1 | 66.64% | 87.09% | 1.97M | (5.93×) | 395.40M | (4.59×) |
| FALCON-k2 | 68.03% | 88.26% | 3.22M | (3.63×) | 653.00M | (2.78×) |
| FALCON-k3 | 69.07% | 88.56% | 4.48M | (2.61×) | 910.61M | (1.99×) |

method which stores only assignments and centroids of weights. While using the model, weights are loaded according to assignments and centroids. Pruning (Han et al. (2014); Li et al. (2016)) aims at removing useless weights or setting them to zero. Although weight-sharing and pruning can significantly reduce the model size, they are not efficient in reducing the amount of computation. Quantizing (Courbariaux et al. (2015; 2016); Hou et al. (2017); Zhu et al. (2017)) the model into binary or ternary weights reduces model size and computation simultaneously: replacing arithmetic operations with bit-wise operations remarkably accelerates the model.

Layer-wise approaches are also employed to efficiently compress models. A typical example of such approaches is low-rank approximation (Lebedev et al. (2015); Kim et al. (2016b); Novikov et al. (2015)); it treats the weights as a tensor and uses general tensor approximation methods to compress the tensor. To reduce computation, approximation methods should be carefully chosen, since some of approximation methods may increase computation of the model.

Compressing existing models has limitations since they are originally designed to be deep and large to give high accuracy. A recent trend is to design a brand new architecture that is small and efficient. Mobilenet (Howard et al. (2017)), MobilenetV2 (Sandler et al. (2018)), Shufflenet (Zhang et al. (2017)), and ShufflenetV2 (Ma et al. (2018)) are the most representative approaches, and they use depthwise convolution and pointwise convolution as building blocks for designing convolution layers. Our proposed FALCON gives a thorough interpretation of depthwise convolution and pointwise convolution, and applies them into model compression, giving the best accuracies with regard to compression and computation.

## 6 CONCLUSION

We propose FALCON, an accurate and lightweight convolution method to replace standard convolution. By interpreting existing convolution methods based on depthwise separable convolution using EHP operation, FALCON and its general version rank-$k$ FALCON provide accurate and efficient compression on CNN. We also propose FALCON-branch, a variant of FALCON integrated into a branch architecture of CNN for model compression. Extensive experiments show that FALCON and its variants give the best accuracy for a given number of parameter or computation, outperforming other convolution models based on depthwise separable convolution. Compared to the standard convolution, FALCON and FALCON-branch give up to 8× compression and 8× computation reduction while giving similar accuracy. We also show that rank-$k$ FALCON provides better accuracy than the standard convolution, while using smaller numbers of parameters and computations.

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

Table 4: Symbols.

| Symbol | Description |
|--------|-------------|
| $\mathcal{K}$ | convolution kernel of size $\mathbb{R}^{D \times D \times M \times N}$ |
| $\mathcal{I}$ | input feature maps of size $\mathbb{R}^{H \times W \times M}$ |
| $\mathcal{O}$ | output feature maps of size $\mathbb{R}^{H' \times W' \times N}$ |
| $D$ | height and width of kernel (kernel size) |
| $M$ | number of input feature map (input channels) |
| $N$ | number of output feature map (output channels) |
| $H$ | height of input feature map |
| $W$ | width of input feature map |
| $H'$ | height of output feature map |
| $W'$ | width of output feature map |
| $s$ | stride |
| $p$ | padding |
| $\odot_{p,q}$ | Extended Hadamard Product (EHP) |
| $t$ | expansion ratio in MobilenetV2 |
| $g$ | number of groups in Shufflenet |

## A  CONVOLUTIONAL NEURAL NETWORK

Convolutional Neural Network (CNN) is a type of deep neural network used mainly for structured data. CNN uses convolution operation in convolution layers. In the following, we discuss CNN when applied to typical image data with RGB channels.

Each convolution layer has three components: input feature maps, convolution kernel, and output feature maps. The input feature maps $\mathcal{I} \in \mathbb{R}^{H \times W \times M}$ and the output feature maps $\mathcal{O} \in \mathbb{R}^{H' \times W' \times N}$ are 3-dimensional tensors, and the convolution kernel $\mathcal{K} \in \mathbb{R}^{D \times D \times M \times N}$ is a 4-dimensional tensor.

The convolution operation is defined as:

$$\mathcal{O}_{h',w',n} = \sum_{i=1}^{D} \sum_{j=1}^{D} \sum_{m=1}^{M} \mathcal{K}_{i,j,m,n} \cdot \mathcal{I}_{h_i,w_j,m} \tag{3}$$

where the relations between height $h_i$ and width $w_j$ of input, and height $h'$ and width $w'$ of output are as follows:

$$h_i = (h' - 1)s + i - p \quad \text{and} \quad w_j = (w' - 1)s + j - p \tag{4}$$

where $s$ is the stride size, and $p$ is the padding size. The third and the fourth dimensions of the convolution kernel $\mathcal{K}$ must match the number $M$ of input channels, and the number $N$ of output channels, respectively.

Convolution kernel $\mathcal{K}$ can be seen as $N$ 3-dimensional filters $\mathcal{F}_n \in \mathbb{R}^{D \times D \times M}$. Each filter $\mathcal{F}_n$ in kernel $\mathcal{K}$ performs convolution operation while sliding over all spatial locations on input feature maps. Each filter produces one output feature map.

# B  PROOFS OF THEOREMS

## B.1  PROOF OF THEOREM 1

*Proof.* From the definition of EHP, $\mathcal{K}_{i,j,m,n} = \mathcal{D}_{i,j,m} \cdot \mathbf{P}_{m,n}$. Based on equation 3, we replace the kernel $\mathcal{K}_{i,j,m,n}$ with the depthwise convolution kernel $\mathcal{D}_{i,j,m}$ and the pointwise convolution kernel $\mathbf{P}_{m,n}$.

$$\mathcal{O}_{h',w',n} = \sum_{i=1}^{D} \sum_{j=1}^{D} \sum_{m=1}^{M} \mathcal{D}_{i,j,m} \cdot \mathbf{P}_{m,n} \cdot \mathcal{I}_{h_i,w_j,m}$$

where $\mathcal{I}_{h_i,w_j,m}$ is the $(h_i, w_j, m)$-th entry of the input. We split the above equation into the following two equations.

$$\mathcal{O}'_{h',w',m} = \sum_{i=1}^{D} \sum_{j=1}^{D} \mathcal{D}_{i,j,m} \cdot \mathcal{I}_{h_i,w_j,m} \tag{5}$$

$$\mathcal{O}_{h',w',n} = \sum_{m=1}^{M} \mathbf{P}_{m,n} \cdot \mathcal{O}'_{h',w',m} \tag{6}$$

where $\mathcal{O}'_{h',w',m} \in \mathbb{R}^{H' \times W' \times M}$ is an intermediate tensor. Note that equation 5 and equation 6 correspond to the depthwise convolution and the pointwise convolution, respectively. Therefore, the output $\mathcal{O}'_{h',w',m}$ is equal to the output applying depthwise separable convolution used in Mobilenet. □

## B.2  PROOF OF THEOREM 2

*Proof.* From equation 3, we replace the kernel $\mathcal{K}_{i,j,m,n}$ with the pointwise convolution kernel $\mathbf{P}$ and the depthwise convolution kernel $\mathcal{D}$.

$$\mathcal{O}_{h',w',n} = \sum_{m=1}^{M} \sum_{i=1}^{D} \sum_{j=1}^{D} \mathbf{P}_{m,n} \cdot \mathcal{D}_{i,j,n} \cdot \mathcal{I}_{h_i,w_j,m}$$

where $\mathcal{I}_{h_i,w_j,m}$ is the $(h_i, w_j, m)$-th entry of the input $\mathcal{I}$. We split the above equation into the following two equations.

$$\mathcal{O}'_{h_i,w_j,n} = \sum_{m=1}^{M} \mathbf{P}_{m,n} \cdot \mathcal{I}_{h_i,w_j,m} \tag{7}$$

$$\mathcal{O}_{h',w',n} = \sum_{i=1}^{D} \sum_{j=1}^{D} \mathcal{D}_{i,j,n} \cdot \mathcal{O}'_{h_i,w_j,n} \tag{8}$$

where $\mathcal{I}$, $\mathcal{O}'$, and $\mathcal{O}$ are the input, the intermediate, and the output tensors of convolution layer, respectively. Note that equation 7 and equation 8 correspond to pointwise convolution and depthwise convolution, respectively. Therefore, the output $\mathcal{O}_{h',w',n}$ is equal to the output applying FALCON. □

## C    QUANTITATIVE ANALYSIS

In this section, we evaluate the compression and the computation reduction of FALCON and rank-$k$ FALCON. All the analysis is based on one convolution layer. The comparison of the numbers of parameters and FLOPs of FALCON and other competitors is in Appendix G.

### C.1    FALCON

We analyze the compression and the computation reduction rates of FALCON in Theorems 3 and 4.

**Theorem 3.** *Compression Rate (CR) of* FALCON *is given by*

$$CR = \frac{\text{\# of parameters in standard convolution}}{\text{\# of parameters in FALCON}} = \frac{D^2 MN}{MN + D^2 N}$$

*where $D^2$ is the size of standard kernel, $M$ is the number of input channels, and $N$ is the number of output channels.*

*Proof.* Standard convolution kernel has $D^2 MN$ parameters. FALCON includes pointwise convolution and depthwise convolution which requires $MN$ and $D^2 N$ parameters, respectively. Thus, the compression rate of FALCON is $CR = \dfrac{D^2 MN}{MN + D^2 N}$. $\qquad\square$

**Theorem 4.** *Computation Reduction Rate (CRR) of* FALCON *is described as:*

$$CRR = \frac{\text{\# of FLOPs in standard convolution}}{\text{\# of FLOPs in FALCON}}$$
$$= \frac{H'W'MD^2N}{HWMN + H'W'D^2N}$$

*where $H'$ and $W'$ are the height and the width of output, respectively, and $H$ and $W$ are the height and the width of input, respectively.*

*Proof.* The standard convolution operation requires $H'W'D^2MN$ FLOPs (Molchanov et al. (2017)). FALCON includes pointwise convolution and depthwise convolution. Pointwise convolution has kernel size $D = 1$ with stride $s = 1$ and no padding, so the intermediate tensor $\mathcal{O}'$ has the same height and width as those of the input feature maps. Thus, pointwise convolution needs $HWMN$ FLOPs. Depthwise convolution has the number of input channel $M = 1$, so it needs $H'W'D^2N$ FLOPs. The total FLOPs of FALCON is $HWMN + H'W'D^2N$, thus the computation reduction rate of FALCON is $CRR = \dfrac{H'W'D^2MN}{HWMN + H'W'D^2N}$. $\qquad\square$

### C.2    RANK-K FALCON

We analyze the compression and computation reduction rates of rank-$k$ FALCON in Theorem 5.

**Theorem 5.** *Compression Rate ($CR_k$) and Computation Reduction Rate ($CRR_k$) of rank-k* FALCON *are described as:*

$$CR_k = \frac{CR}{k} \qquad CRR_k = \frac{CRR}{k}$$

*Proof.* The numbers of parameters and FLOPs increase for k times since rank-k FALCON duplicates FALCON for k times. Thus, the compression rate and the computation reduction rate are calculated as $CR_k = \dfrac{CR}{k}$ and $CRR_k = \dfrac{CRR}{k}$. $\qquad\square$

## D    DESCRIPTION OF RELATED CONVOLUTION UNITS

**MobileNetV2.**   Sandler et al. (2018) proposed a new convolution architecture which we call as MobileConvV2, in their MobilenetV2 model. MobileConvV2 consists of three sub-layers as shown in Figure 5(b). The first and the third sub-layers are pointwise convolution for adjusting the number of channels. The first sub-layer expands the number of channels from $M$ to $tM$, where $t$ is an

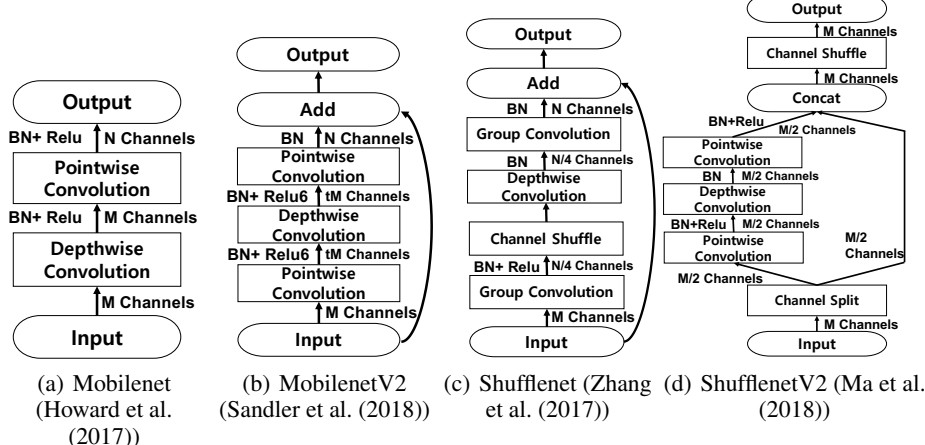

(a) Mobilenet (Howard et al. (2017))   (b) MobilenetV2 (Sandler et al. (2018))   (c) Shufflenet (Zhang et al. (2017))   (d) ShufflenetV2 (Ma et al. (2018))

Figure 5: Comparison of architectures based on depthwise separable convolution. BN denotes batch-normalization, and Relu is an activation function.

expansion ratio. The second sub-layer is a $D \times D$ depthwise convolution. Since depthwise convolution cannot change the number of channels, the third sub-layer adjusts the number of channels from $tM$ to $N$. There is a shortcut connection between the input, and the output of the third sub-layer to facilitate flow of gradient across multiple layers. MobileConvV2 needs $tM^2 + D^2tM + tMN$ parameters and $tHWM^2 + tH'W'D^2M + tH'W'MN$ FLOPs.

**ShuffleNet.** Zhang et al. (2017) proposed a computation-efficient CNN architecture named Shufflenet. As shown in Figure 5(c), each unit of Shufflenet (we call it ShuffleUnit) consists of three sub-layers, first group pointwise convolution, depthwise convolution, and second group pointwise convolution, as well as a shortcut. The number of depthwise convolution channels is $\frac{1}{4}$ of output channels $N$. ShuffleUnit uses group convolution in two pointwise convolution layers to reduce the parameters and FLOPs. However, it is hard to exchange information among groups when group convolutions are stacked. To deal with this problem, ShuffleUnit adds a channel shuffle layer after the first pointwise group convolution. The channel shuffle layer rearranges the order of channels. making it possible to obtain information from different groups. The number of groups is represented as $g$. ShuffleUnit needs $\frac{1}{4g}MN + \frac{1}{4}D^2N + \frac{1}{4g}N^2$ parameters and $\frac{1}{4g}HWMN + \frac{1}{4}H'W'D^2N + \frac{1}{4g}H'W'N^2$ FLOPs.

**ShufflenetV2.** Ma et al. (2018) proposed a practically efficient CNN architecture ShufflenetV2. As shown in Figure 5(d), each unit of ShufflenetV2 (we call it ShuffleUnitV2) consists of two branches. The left branch consists of two pointwise convolutions and one depthwise convolution like MobileConvV2, and the right branch is an identity operation. Note that outputs of both branches maintain the number of channels as $M/2$. The final output is produced by concatenating and shuffling the output tensors from both of the branches. ShuffleUnitV2 needs $\frac{1}{2}(M^2 + D^2M)$ parameters and $\frac{1}{2}HW(M^2 + D^2M)$ FLOPs.

## E  GENERALITY OF EHP

We show that EHP is a key operation to understand other convolution architectures based on depthwise separable convolution.

**MobilenetV2.** As shown in Figure 5(b), MobilenetV2 has an additional pointwise convolution before depthwise convolution in Mobilenet: one layer of MobilenetV2 consists of two pointwise convolutions and one depthwise convolution. In another point of view, MobilenetV2 can be understood as FALCON followed by additional pointwise convolution; i.e., MobilenetV2 performs EHP operation as FALCON does, and performs additional pointwise convolution after that.

**Shufflenet.** As shown in Figure 5(c), Shufflenet consists of depthwise convolution and pointwise group convolution which is a variant of pointwise convolution. We represent the convolution layer of Shufflenet using EHP as follows. Let $g$ be the number of groups. We divide the standard convolution

kernel $\mathcal{K} \in \mathbb{R}^{D \times D \times M \times N}$ into $g$ group standard convolution kernels. Then, the relation of $g$-th group standard convolution kernel $\mathcal{K}^g \in \mathbb{R}^{D \times D \times \frac{M}{g} \times \frac{N}{g}}$ with regard to $g$-th depthwise convolution kernel $\mathcal{D}^g \in \mathbb{R}^{D \times D \times \frac{M}{g}}$ and $g$-th pointwise group convolution kernel $\mathbf{P}^g \in \mathbb{R}^{\frac{M}{g} \times \frac{N}{g}}$ is

$$\mathcal{K}^g = \mathcal{D}^g \odot_E \mathbf{P}^g \quad \text{s.t.} \quad \mathcal{K}^g_{i,j,m_g,n_g} = \mathcal{D}^g_{i,j,m_g} \cdot \mathbf{P}^g_{m_g,n_g}$$

where $m_g = 1, 2, ..., \frac{M}{g}$ and $n_g = 1, 2, ..., \frac{N}{g}$. Each group standard convolution is equivalent to the combination of a depthwise convolution and a pointwise convolution, and thus easily expressed with EHP as in Mobilenet.

Therefore, each layer of Shufflenet is equivalent to the layer consisting of one group convolution followed by standard convolution.

**ShufflenetV2.** As shown in Figure 5(d), the left branch of ShufflenetV2 has the same convolutions as in MobilenetV2: it consists of two pointwise convolutions and one depthwise convolution. Like MobilenetV2, the left branch of ShufflenetV2 can be understood as FALCON followed by additional pointwise convolution.

# F    FITTING OTHER CONVOLUTION UNITS INTO MODELS

**DSConv.** DSConv (shown in Figure 5(a)) has the most similar architecture as FALCON among competitors, and thus DSConv has nearly the same number of parameters as that of FALCON. As in the setting of FALCON, the existence of BN and ReLU at the end of DSConv depends on that of StConv.

**MobileConvV2.** In MobileConvV2 architecture (shown in Figure 5(b)), we adjust the numbers of parameters and FLOPs by changing the expansion ratio $t$ as described in Appendix D, which is represented as 'MobileConvV2-$t$'. We choose $t = 0.5$ as the baseline MobileConvV2 to compare with FALCON, since two pointwise convolutions bring lots of parameters and FLOPs to MobileConvV2.

**ShuffleUnit.** In ShuffleUnit (shown in Figure 5(c)), we adjust the numbers of parameters and FLOPs by changing the width multiplier $\alpha$ (Howard et al. (2017)) and the number of groups $g$, which is represented as 'ShuffleUnit $\alpha \times$(g=$g$)'. Note that the width multiplier is used to adjust the number of input channels $M$ and the number of output channels $N$ of a convolution layer; if the width multiplier is $\alpha$, the numbers of input and output channels become $\alpha M$ and $\alpha N$, respectively. While experimenting with ResNet, we find that ShuffleUnit does not cooperate well with ResNet: ResNet34 with ShuffleUnit does not converge. We suspect that residual block and ShuffleUnit may conflict with each other because of redundant residual connections: the gradient may not find the right path towards previous layers. For this reason, we delete the shortcut of all residual blocks in ResNet34 when using ShuffleUnit.

**ShuffleUnitV2.** In ShuffleUnitV2 (shown in Figure 5(d)),we also adjust the number of parameters and FLOPs by changing the width multiplier $\alpha$, which is represented as 'ShuffleUnitV2 $\alpha \times$'. Other operations of ShuffleUnitV2 stay the same as in Ma et al. (2018).

# G    PARAMETERS AND FLOPS

We summarize the numbers of parameters and FLOPs for FALCON and competitors in Table 5.

Table 5: the numbers of parameters and FLOPs of FALCON and competitors. Symbols are described in Table 4.

| Convolution | # of parameters | # of FLOPs |
|---|---|---|
| FALCON | $MN + D^2N$ | $HWMN + H'W'D^2N$ |
| FALCON-branch | $\frac{1}{4}M^2 + \frac{1}{2}D^2M$ | $\frac{1}{4}HWM^2 + \frac{1}{2}HWD^2M$ |
| DSConv | $MN + D^2M$ | $HWD^2M + H'W'MN$ |
| MobilenetV2 | $tM^2 + tD^2M + tMN$ | $tHWM^2 + tH'W'D^2M + tH'W'MN$ |
| Shufflenet | $\frac{1}{4}(\frac{MN}{g} + D^2N + \frac{N^2}{g})$ | $\frac{1}{4}(\frac{HWMN}{g} + H'W'D^2N + \frac{H'W'N^2}{g})$ |
| ShufflenetV2 | $\frac{1}{2}(M^2 + D^2M)$ | $\frac{1}{2}HW(M^2 + D^2M)$ |
| StConv-branch | $\frac{1}{4}D^2M^2$ | $\frac{1}{4}HWD^2M^2$ |
| Standard convolution | $D^2MN$ | $H'W'D^2MN$ |

