# OpenReview forum: "FALCON: Fast and Lightweight Convolution for Compressing and Accelerating CNN"
_ICLR.cc/2020/Conference — Reject_

### Official Review · AnonReviewer2 · 2019-10-23
**Official Blind Review #2**

**Rating:** 6

**Review:**

The paper proposes a CNN compression method, based on the so called EHP operation, which can be used to  analyze and generalize depthwise separable convolution. Based on EHP, the paper develops depthwise separable convolution to compress CNNs, and extend it to a rank-k approach with further improved accuracy. Some analysis is provided about the operation equivalence. The experiments on standard benchmark datasets show the effectiveness of the method.

Probably the most important contribution of this work is that it proposes a new operation that can summarize/generalize the existing depthwise separable convolution and reveal the relationship with the standard convolution. The work is meaningful to me because a compact representation of CNNs can largely reduce the computational resources and storage.
The experiments are extensive as well.

**Experience Assessment:**

I do not know much about this area.

**Review Assessment: Checking Correctness Of Derivations And Theory:**

I assessed the sensibility of the derivations and theory.

**Review Assessment: Checking Correctness Of Experiments:**

I assessed the sensibility of the experiments.

**Review Assessment: Thoroughness In Paper Reading:**

I made a quick assessment of this paper.

---

> ### Author Response · Authors · 2019-11-15
> **Response to Reviewer #2**
>
> We thank the reviewer for the comments in our paper. As the reviewer mentioned, we agree that it is important to design efficient convolutional neural networks without loss of performance. To address this issue, we propose a mathematical formulation to generalize depthwise separable convolution, and design an efficient and accurate CNN architecture by exploiting the formulation.

---

### Official Review · AnonReviewer1 · 2019-10-24
**Official Blind Review #1**

**Rating:** 6

**Review:**

Overview:
The paper is dedicated to studying fast and lightweight convolution for efficient compression and retaining original accuracy. In this paper, the authors interpret existing convolution methods based on depthwise separable convolution and derive FALCON. They claim their FALCON mathematically approximate the standard convolution kernel and achieves a better TA/efficiency tradeoff. They conduct extensive experiments to show that FALCON based method 1) outperforms previous state-of-the-art methods; 2) achieve 8X efficiency while ensuring similar TA.

Strength Bullets:
1. The authors give a detailed interpretation of the depthwise separable convolution method via EHP. Then they propose novel FALCON has better accuracy than competitors while having similar compression and computation reduction rates.
2. The paper is well organized and easy to read. The author conducts extensive experiments to show FALCON based method not only surpass other depthwise separable convolution models but also give up to 8 times efficiency while giving similar accuracy.

Weakness Bullets:
1. The authors need to provide more ablation experiments for two components of FALCON: 1) align depthwise and pointwise convolution; 2) initialize kernels. Moreover, for the order of performing depthwise convolution and pointwise convolution, it also needs experimental support.
2. I would like to see a comparison of the compression rate with previous filter compression methods. i.e, Soft weight-sharing (ICLR'17), Deep K-means (ICML'18) and so on.
3. [Minor] Even if there is computation reduction (Flops), it is not interesting enough. I am very curious about how much efficiency gains (i.e. latency) in the software implementation of FALCON.

Recommendation:
Although there are a few flaws in experiment design, it still is a novel and good technique. This is a weak accept.

**Experience Assessment:**

I have published in this field for several years.

**Review Assessment: Checking Correctness Of Derivations And Theory:**

I carefully checked the derivations and theory.

**Review Assessment: Checking Correctness Of Experiments:**

I carefully checked the experiments.

**Review Assessment: Thoroughness In Paper Reading:**

I read the paper thoroughly.

---

> ### Author Response · Authors · 2019-11-15
> **Response to Reviewer #1**
>
> I sincerely appreciate your in-depth review. We address your comments and attach additional experimental results.
>
> Q. (Ablation study for depthwise and pointwise convolutions) The authors need to provide more ablation experiments for two components of FALCON: 1) align depthwise and pointwise convolution; 2) initialize kernels. Moreover, for the order of performing depthwise convolution and pointwise convolution, it also needs experimental support.
>
> A. We ran additional experiments for ablation study, and reported the results for FALCON without initialization in Table 2 of the revised paper.
> 1. (Align depthwise and pointwise convolution) For the order of performing depthwise convolution and pointwise convolution, we compared FALCON with EHP-in in Table 2 of the revised paper. FALCON applies depthwise convolution after pointwise convolution while EHP-in applies pointwise convolution after depthwise convolution. The order of depthwise and pointwise convolutions improves the accuracy by up to 5%.
> 2. (Kernel initialization) For the effectiveness of initialization, we compared the accuracy difference between FALCON without initialization and FALCON with initialization. The initialization technique improves the accuracy of FALCON by up to 0.5%. We also note that the initialization does not always improve the performance since depthwise separable convolutions alone can learn rich representations from scratch pretty well.
> As a result, the overall performance is more sensitive to the ordering of depthwise separable convolution than to initialization.
>
> Q. (Comparison between other compression techniques) I would like to see a comparison of the compression rate with previous filter compression methods. i.e, Soft weight-sharing, Deep K-means and so on.
>
> A. For CIFAR10, CIFAR100, SVHN datasets, we ran additional experiments to compare FALCON with Tucker decomposition technique. We compress standard convolution kernels using Tucker decomposition and set the number of parameters and flops to be approximately equal to those of FALCON. For all datasets, our experimental results show that FALCON performs better than Tucker decomposition. For CIFAR100, there is a huge accuracy gap between standard convolution and Tucker decomposition since 1) Tucker decomposition cannot approximate the standard convolution well enough, and 2) the representational power of the convolution kernels resulting from Tucker decomposition is not sufficient to learn complex representations of CIFAR100. Note that CIFAR100 is a more complex dataset since the number of classes of CIFAR100 is much greater than those of CIFAR10 and SVHN, while the number of training instances is similar to those of CIFAR10 and SVHN. The fact that the accuracy gap between the standard convolution and FALCON is smaller than that between the standard convolution and Tucker decomposition demonstrates that FALCON has greater representational efficiency for the same number of parameters and flops.
>
>                | VGG19-cifar10 |ResNet34-cifar10 |VGG19-cifar100 |ResNet34-cifar100 |VGG19-svhn |ResNet-svhn
> StConv  |      93.56 %         |          94.01%          |         72.10%        |            73.94%          |       95.28%     |      95.09%
> FALCON|      93.40%          |          92.78%          |         71.63%        |            71.83%          |       95.48%     |      94.83%
> Tucker  |       90.91%         |          92.20%          |         57.32%        |            58.1%            |      95.30%      |      91.22%
>
> Note that we compare FALCON with depthwise separable convolution-based methods. The reasons are 1) we are more interested in how to improve utilization of depthwise separable convolution which recent state-of-the-art methods including MobileNet and ShuffleNet are based on, and 2) other methods such as soft weight-sharing and deep K-means reduce only the number of parameters but do not reduce the number of computations, while FALCON reduces both, so we wanted to consider only competitors that also reduce both the number of parameters and the number of computations.
>
> Q. [Minor point] (Running time) Even if there is computation reduction (Flops), it is not interesting enough. I am very curious about how much efficiency gains (i.e. latency) in the software implementation of FALCON.
>
> A. The reason why we reported computation reduction, rather than latency, is that latency is affected by many factors, such as batch size, image size, platforms (GPU, ARM), implementation of convolution, etc. Some of the factors are inaccessible while using Pytorch framework. For example, training and inference are done on GPU with cuDNN. While the latest cuDNN library on GPU is specially optimized for 3×3 convolution, it is not optimized for depthwise convolution. In this situation, it is hard to make a fair comparison of latency among different models. We implemented FALCON on convolution level using Pytorch framework, but we didn’t optimize FALCON on CUDA level.

---

### Official Review · AnonReviewer3 · 2019-10-28
**Official Blind Review #3**

**Rating:** 3

**Review:**

This paper proposed a model compression method: Falcon and rank-k Falcon. Both are used to compress CNN type of models by replacing standard convolution layer with a compact Falcon or rank-k Falcon layer to compress the model. Falcon's main idea is to decompose the traditional convolution kernel K into two smaller tensors, one is depthwise convolution kernel D and pointwise convolution kernel P. And DP will reconstruct the original kernel K. Since D+P's memory is  D*D*M+N*M which is smaller than the original size D*D*M*N, and thus when N is large, the memory saving could be large. The paper is in general in good writing and very easy to read.

Below I have several concerns/suggestions for this paper:

1: Novelty. What is the main difference between this method and all the other tensor decomposition based methods for CNN compression? There are so many tensor decomposition based methods for CNN, and seems Falcon belongs to one of them. The one (maybe) special for Falcon is that it only decomposes along one dimension. Why this method could perform better than other tensor based decomposition methods(some of them are having even smaller memory footprint as they decompose more dimensions) or Falcon could be one special case of it?

2: In Section 3.3 and 3.4, the proposed Falcon and rank-k Falcon seems is fully recovering the original K, see equations above Theorem 2 and above Section 3.5, should it be minimizing the reconstruction error as other tensor decomposition methods? And how to find the solutions P and D from K? how the model is getting trained if using Falcon or rank-k Falcon? Do you have retrain step after the decomposition of K?

3: In the experiment, no any other standard compression techniques such as quantization, low-rank, weight-sharing, sparse, etc are compared.  This makes us curious about the benefit of the proposed methods over other methods.

In summary, I am mostly worried about the novelty of the paper, and wondering how the model is getting trained, and the comparison with other compression techniques.

**Experience Assessment:**

I have published in this field for several years.

**Review Assessment: Checking Correctness Of Derivations And Theory:**

I assessed the sensibility of the derivations and theory.

**Review Assessment: Checking Correctness Of Experiments:**

I assessed the sensibility of the experiments.

**Review Assessment: Thoroughness In Paper Reading:**

I read the paper thoroughly.

---

> ### Author Response · Authors · 2019-11-15
> **Response to Reviewer #3**
>
> We would like to express our sincere gratitude for the high-quality reviews and comments. Attached below are our responses to your suggestions and concerns.
>
> Q. Why this method could perform better than other tensor based decomposition methods (some of them are having even smaller memory footprint as they decompose more dimensions) or Falcon could be one special case of it?
>
> A. Our formulation of the relation between depthwise separable convolutions and standard convolution allows FALCON to be the only method which takes advantages of both tensor decomposition methods and depthwise separable convolution-based methods. FALCON 1) provides a good starting point to recover the accuracy by approximating the standard convolution (advantage of tensor decomposition), 2) preserves the representational power (advantage of depthwise separable convolution), and 3) reduces both the numbers of parameters and flops. This is why FALCON performs better than other tensor decomposition methods and depthwise separable convolution based methods. Although the depthwise separable convolution technique has been widely used for designing efficient convolutional neural networks [1,2,3,4], those methods do not formulate the relation between standard convolutions and depthwise separable convolutions. Therefore, previous works based on depthwise separable convolutions do not transfer the knowledge of the original standard convolution into the pointwise and depthwise convolution kernels.
>
> Q. The main difference between this method and all the other tensor decomposition based methods for CNN compression.
>
> A. The main difference is that FALCON not only decomposes a standard convolution kernel into two tensors, which correspond to pointwise and depthwise convolution kernels, but also decouples tasks done by a standard convolution kernel so that each of the two decomposed kernels independently performs a different task. A depthwise convolution extracts spatial features and a pointwise convolution merges features along the channel dimension, while a standard convolution simultaneously performs the two tasks. It allows FALCON to achieve representational efficiency and reduce the number of parameters and flops at the same time. Decoupling the tasks preserves the representation power of the network while reducing the number of parameters. On the other hand, previous tensor decomposition methods such as cp and Tucker decomposition reduce the number of flops and parameters by decomposing a standard convolution kernel into several small standard convolution kernels. These methods cause a decrease in the representation power of a network since small decomposed convolutions do redundant feature extraction, and the size of the channel dimension of intermediate tensors are small. Note that the size of the channel dimension of intermediate tensors is equal to the rank of the decomposition, and is smaller than the size of the channel dimension of input and output tensors.
>
> Q. (Initialization and training process) should it be minimizing the reconstruction error as other tensor decomposition methods? And how to find the solutions P and D from K? how the model is getting trained if using Falcon or rank-k Falcon? Do you have retrain step after the decomposition of K?
>
> A. We take three steps to train the model with FALCON. We 1) randomly initialize a depthwise convolution kernel D and a pointwise convolution kernel P, and 2) update them to minimize the reconstruction error, and 3) finetune the model with them. In step 2, we update the pointwise convolution kernel P and the depthwise convolution kernel D using gradient descent such that the mean squared error of their EHP product and the standard convolution kernel is minimized. It is described in Section 4.2 of the paper.
>
> References
> [1] Chollet and Franois. Xception: Deep learning with depthwise separable convolutions, 2016. URL http://arxiv.org/abs/1610.02357.
> [2] Ningning et al. Shufflenet V2: practical guidelines for efficient CNN architecture design, 2018. URL http://arxiv.org/abs/1807.11164.
> [3] Mark Sandler et al. Mobilenetv2: Inverted residuals and linear bottlenecks, 2018. URL https://arxiv.org/abs/1801.04381.
> [4] Xiangyu Zhang et al. Shufflenet: An extremely efficient convolutional neural network for mobile devices, 2017. URL http://arxiv.org/abs/1707.01083.

---

> > ### Author Response · Authors · 2019-11-15
> > **Response to Reviewer #3 (contd.)**
> >
> > Q. (Comparison between other compression techniques) In the experiment, no any other standard compression techniques such as quantization, low-rank, weight-sharing, sparse, etc are compared.
> >
> > A. For CIFAR10, CIFAR100, SVHN datasets, we ran additional experiments to compare FALCON with Tucker decomposition technique. We compress standard convolution kernels using Tucker decomposition and set the number of parameters and flops to be approximately equal to those of FALCON. For all datasets, our experimental results show that FALCON performs better than Tucker decomposition. For CIFAR100, there is a huge accuracy gap between standard convolution and Tucker decomposition since 1) Tucker decomposition cannot approximate the standard convolution well enough, and 2) the representational power of the convolution kernels resulting from Tucker decomposition is not sufficient to learn complex representations of CIFAR100. Note that CIFAR100 is a more complex dataset since the number of classes of CIFAR100 is much greater than those of CIFAR10 and SVHN, while the number of training instances is similar to those of CIFAR10 and SVHN. The fact that the accuracy gap between the standard convolution and FALCON is smaller than that between the standard convolution and Tucker decomposition demonstrates that FALCON has greater representational efficiency for the same number of parameters and flops.
> >                |  VGG19-cifar10 | ResNet34-cifar10 | VGG19-cifar100 | ResNet34-cifar100 | VGG19-svhn | ResNet-svhn
> > StConv  |      93.56 %         |          94.01%          |         72.10%        |            73.94%          |       95.28%     |      95.09%
> > FALCON|      93.40%          |          92.78%          |         71.63%        |            71.83%          |       95.48%     |      94.83%
> > Tucker  |       90.91%          |          92.20%          |         57.32%        |            58.1%            |      95.30%      |      91.22%
> > Note that we compare FALCON with depthwise separable convolution-based methods. The reasons are 1) we are more interested in how to improve utilization of depthwise separable convolution which recent state-of-the-art methods including MobileNet and ShuffleNet are based on, and 2) other methods such as soft weight-sharing and deep K-means reduce only the number of parameters but do not reduce the number of computations, while FALCON reduces both, so we wanted to consider only competitors that also reduce both the number of parameters and the number of computations.

---

### Author Response · Authors · 2019-11-15
**Summary of our revision**

We thank all three reviewers for the high quality comments.
As suggested, we ran additional experiments to revise our paper as follows:
1) we added the results of FALCON without initialization in Table 2 in order to show the effectiveness of the initialization , and
2) updated ablation study part in Section 4.2 to discuss the result on Table 2.

---

### Decision · Program_Chairs · 2019-12-19

**Decision:**

Reject

**Comment:**

The submission presents an approach to accelerating convolutional networks. The framework is related to depthwise separable convolutions. The reviews are split. R3 expresses concerns about the experimental evaluation and results. The AC agrees with these concerns. The AC also notes that the submission is 10 pages long. Taking all factors into account, the AC recommends against accepting the paper.